# Formation Conditions and $^{40}$Ar/$^{39}$Ar Age of the Gem-Bearing Boqueirão Granitic Pegmatite, Parelhas, Rio Grande do Norte, Brazil

**Sabina Strmić Palinkaš [1],\*** , **Ladislav Palinkaš [2]**, **Franz Neubauer [3]** , **Ricardo Scholz [4]**, **Sibila Borojević Šoštarić [5]** and **Vladimir Bermanec [2]**

[1] Department of Geosciences, Faculty of Sciences and Technology, UiT-The Arctic University of Norway in Tromsø, N-9037 Tromsø, Norway
[2] Department of Geology, Faculty of Science, University of Zagreb, Horvatovac 95, HR-10000 Zagreb, Croatia; lpalinkas@geol.pmf.hr (L.P.); vladimir.bermanec@public.carnet.hr (V.B.)
[3] Department of Geography and Geology, Paris-Lodron-University of Salzburg, Hellbrunner Str. 34, A-5020 Salzburg, Austria; franz.neubauer@sbg.ac.at
[4] Departamento de Geologia, Escola de Minas, Universidade Federal de Ouro Preto, Ouro Preto MG-31400-000, Brazil; r_scholz_br@yahoo.com
[5] Faculty of Mining, Geology and Petroleum Engineering, University of Zagreb, Pierottijeva 6, HR-10000 Zagreb, Croatia; sibila.borojevic-sostaric@rgn.hr
\* Correspondence: Sabina.s.palinkas@uit.no; Tel.: +47-77-625-177

**Abstract:** The Boqueirão granitic pegmatite, alias Alto da Cabeça pegmatite, is situated in Borborema Pegmatitic Province (BPP) in Northeast Brazil. This pegmatitic province hosts globally important reserves of tantalum and beryllium, as well as significant quantities of gemstones, including aquamarine, morganite, and the high-quality turquoise-blue "Paraíba Elbaite". The studied lithium-cesium-tantalum Boqueirão granitic pegmatite intruded meta-conglomerates of the Equador Formation during the late Cambrian (502.1 ± 5.8 Ma; $^{40}$Ar/$^{39}$Ar plateau age of muscovite). The pegmatite exhibits a typical zonal mineral pattern with four defined zones (Zone I: muscovite, tourmaline, albite, and quartz; Zone II: K-feldspar (microcline), quartz, and albite; Zone III: perthite crystals (blocky feldspar zone); Zone IV: massive quartz). Huge individual beryl, spodumene, tantalite, and cassiterite crystals are common as well. Microscopic examinations revealed that melt inclusions were entrapped simultaneously with fluid inclusions, suggesting the magmatic–hydrothermal transition. The magmatic–hydrothermal transition affected the evolution of the pegmatite, segregating volatile compounds ($H_2O$, $CO_2$, $N_2$) and elements that preferentially partition into a fluid phase from the viscous silicate melt. Fluid inclusion studies on microcline and associated quartz combined with microthermometry and Raman spectroscopy gave an insight into the P-T-X characteristics of entrapped fluids. The presence of spodumene without other $LiAl(SiO_3)_2$ polymorphs and constructed fluid inclusion isochores limited the magmatic–hydrothermal transition at the gem-bearing Boqueirão granitic pegmatite to the temperature range between 300 and 415 °C at a pressure from 1.8 to 3 kbar.

**Keywords:** gem-bearing pegmatite; fluid inclusions; P-T-X equilibria; spodumene; Ar/Ar dating

## 1. Introduction

Pegmatites, plutonic igneous rocks characterized by extremely coarse crystals with a systematically variable size, represent an important source of industrial minerals (feldspars, quartz, spodumene, petalite), hi-tech mineral commodities (e.g., Li, Cs, Be, Nb, Ta, Sn), and gemstones. The most common pegmatite-hosted gem minerals are colored varieties of beryl (aquamarine, heliodor, and morganite),

Li-rich tourmaline (elbaite-rossmanite and liddicoatite), blue and sherry topaz, transparent varieties of spodumene (kunzite), low-iron spessartine, and optical-grade quartz [1].

The Boqueirão granitic pegmatite, also known as the Alto da Cabeça pegmatite, is situated in the northernmost part of the Serra das Queimadas Mountains, the State of Rio Grande do Norte, Northeast Brazil (Figure 1). The pegmatite is hosted by Borborema Pegmatitic Province (BPP), which represents one of the world's most important sources of tantalum and beryllium, as well as of gemstones, including aquamarine, morganite, and the high-quality turquoise-blue "Paraíba Elbaite" [2–5].

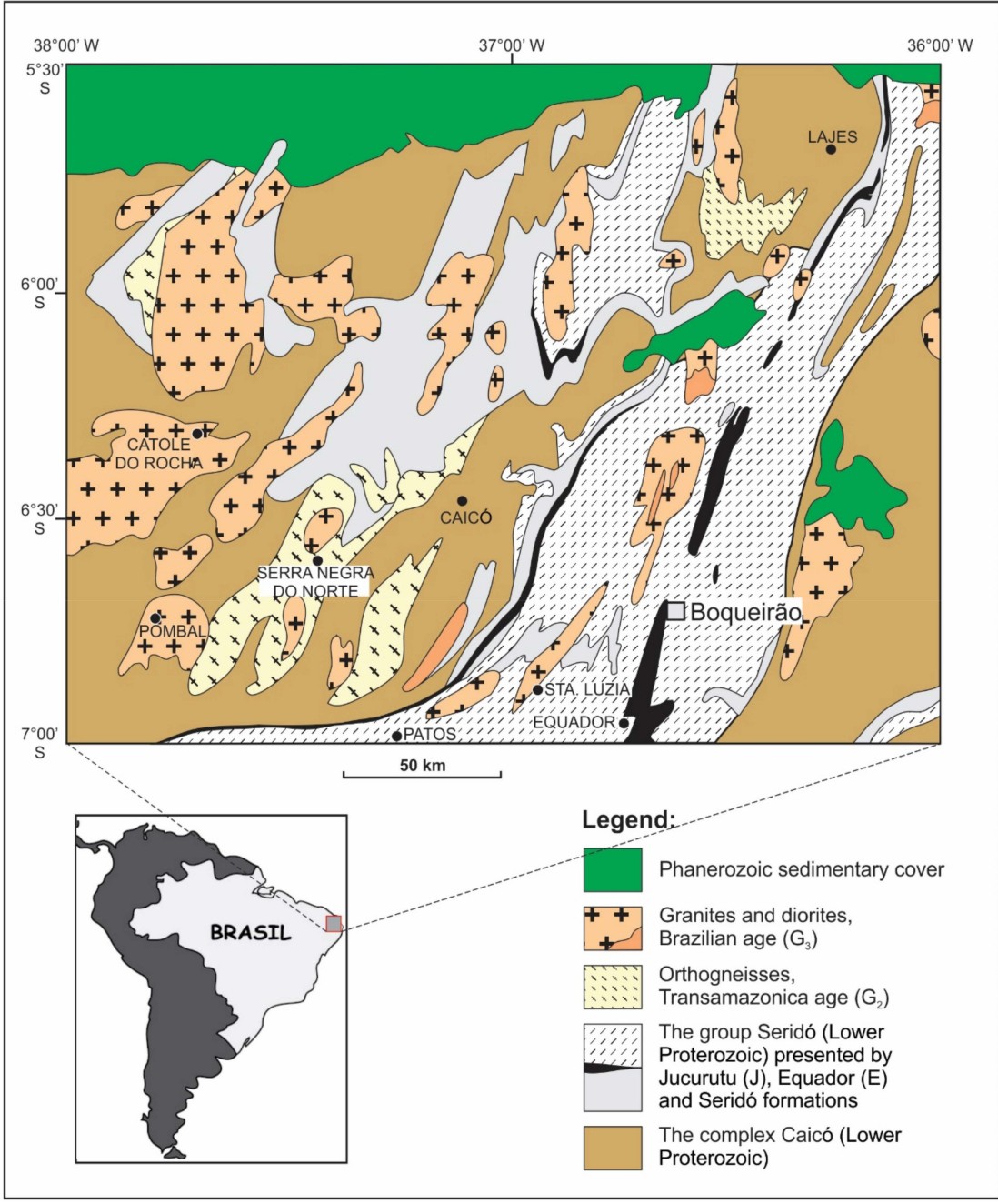

**Figure 1.** A simplified geological map of the Borborema Pegmatitic Province with the location of the Boqueirão granitic pegmatite, Parelhas, Rio Grande do Norte, Brazil. Adapted with permission from [6].

Mineralogical and geochemical features, characterized by a strong enrichment in numerous incompatible elements, including Li, Rb, Cs, Be, Sn, Ta, Nb (with Ta > Nb), B, P, and F, classify the

Boqueirão granitic pegmatite into the lithium-cesium-tantalum (LCT) family [5–9]. According to previously published data, the BPP pegmatites were formed in a relatively narrow range of pressure (2.1 to 4 kbar) but within a wide range of temperatures (390–900 °C) [10–12].

This study brings new fluid inclusion and $^{40}Ar/^{39}Ar$ data from the gem-bearing Boqueirão granitic pegmatite. The fluid inclusion data combined with the calculated thermodynamic equilibria of the established mineral paragenesis shed light on the formation conditions of the Boqueirão granitic pegmatite, whereas the $^{40}Ar/^{39}Ar$ dating confirmed the late Cambrian age of pegmatite emplacement. A particular focus has been given to the recorded magmatic–hydrothermal transition and its potential role in the evolution of the studied pegmatite.

## 2. Geological Setting

The Borborema Pegmatitic Province (6°–7°S, 36°15′–36°45′W) covers the southern part of the Meso- to Neo-Proterozoic Seridó Foldbelt (Figure 1; [13]). The Seridó Foldbelt comprises a basal volcano-sedimentary sequence (Jucurutu Formation), a quartzite–metaconglomerate complex (Equador Formation), and a turbidite–flysch sequence (Seridó Formation). The area is metamorphosed up to the upper amphibolite facies (Abukuma type) and retro-metamorphosed into the upper greenschist-facies grade [14]. Four generations of granite intrusions in the area have been described [15]. The formation of the pegmatites is related to granites of the late- to post-orogenic phase [16], labelled as G4 granites [15,17].

The Boqueirão granitic pegmatite intruded into meta-conglomerates of the Equador Formation [17]. The central part of the pegmatite exhibits a typical zonal mineral distribution. From its margin to the center, the pegmatite consists of (Figure 2): Zone I with comb-textured muscovite and/or tourmaline intergrown with medium-grained albite and quartz (Figure 3a–c); Zone II hosting homogeneous medium-grained K-feldspar (microcline) accompanied by quartz and albite; Zone III composed almost exclusively of large perthite crystals (blocky feldspar zone); and Zone IV, i.e., a monomineralic nucleus of massive milky and/or rose quartz [6,18]. The contact between Zones III and IV was a preferred site for deposition of decimetric to metric, irregular pockets of medium- to fine-grained cleavelandite (albite), muscovite, and lepidolite selvages with some phosphates and disseminated ore minerals (Figure 3c). Individual beryl, spodumene, tantalite, and cassiterite crystals have also been found at the boundary, with the roots of the crystals in the blocky feldspar zone and the tips growing idiomorphically into the former open space, now filled by the massive quartz core (Figure 3d). Occurrences of beryl and tourmaline mineralization at the boundaries between Zones II and III (Figure 3e,f), and spodumene and/or cassiterite in Zones I and II are common ([6] and references therein). The U-Pb dating of manganocolumbite and ferrocolumbite constrained the time of pegmatite emplacement between 509 and 515 Ma [19].

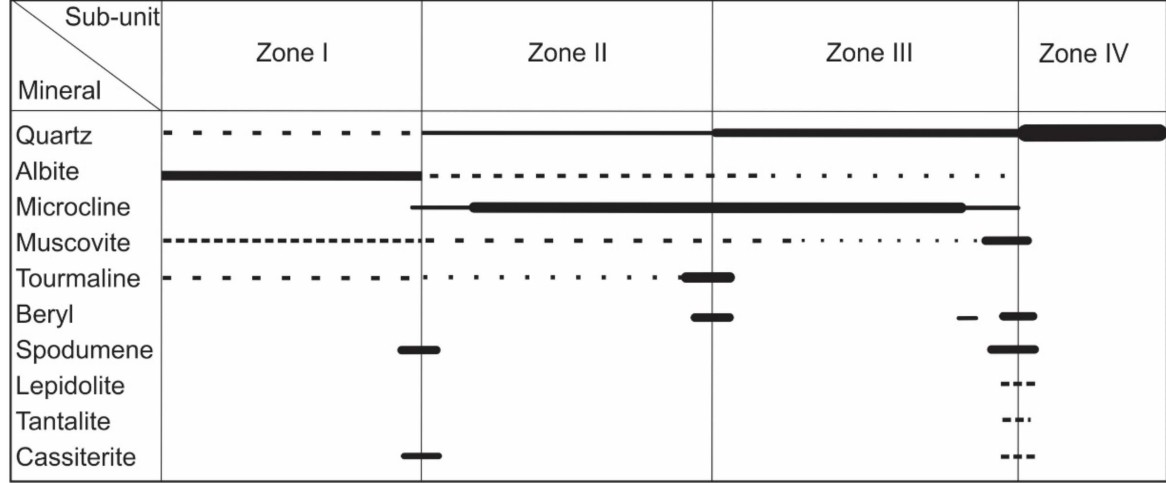

**Figure 2.** The paragenetic sequence of the gem-bearing Boqueirão granitic pegmatite.

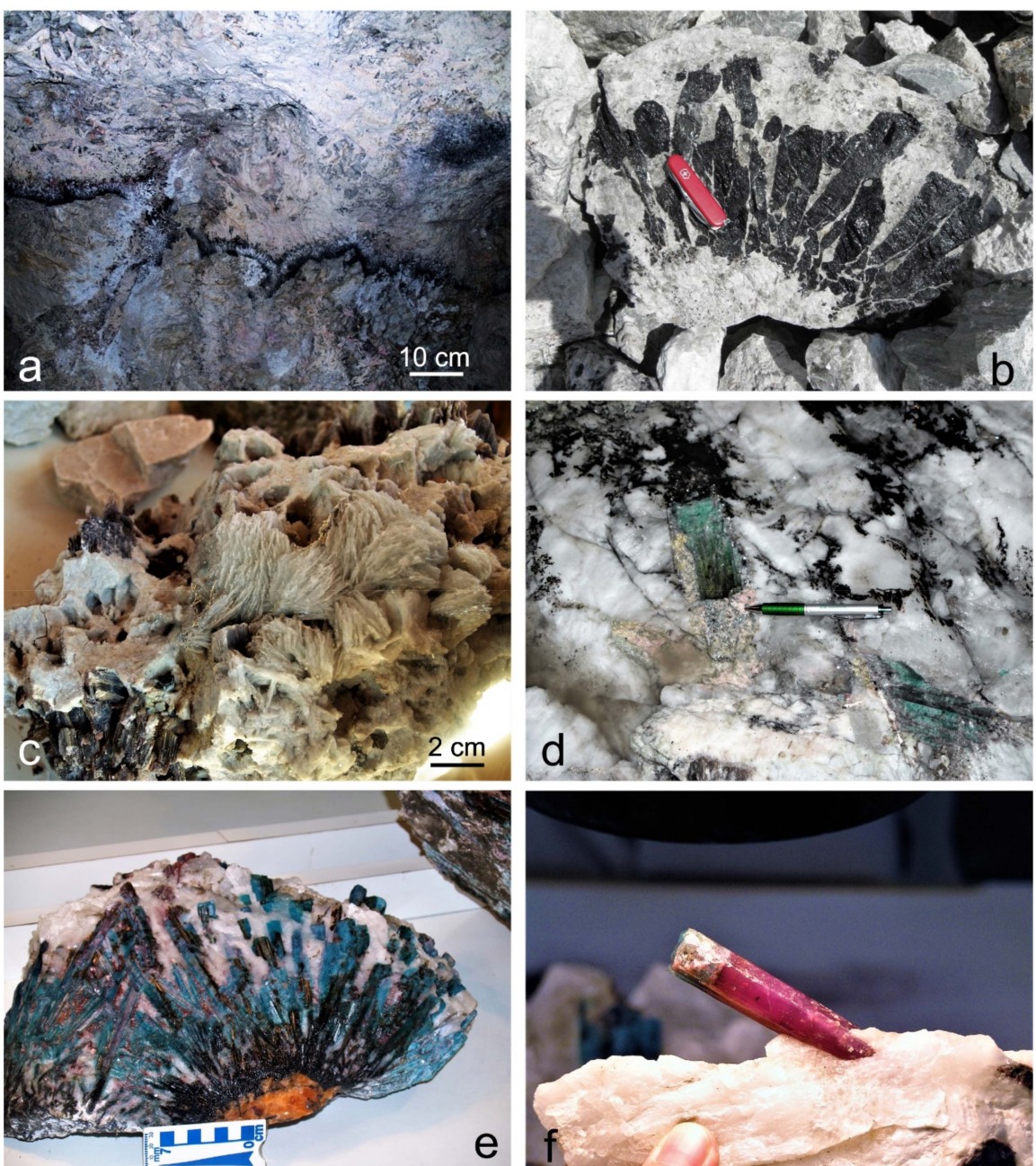

**Figure 3.** Macrophotographs showing: (**a**) A several-centimeters-thick tourmaline front, Zone I; (**b**) Comb-textured tourmaline intergrown within medium-grained albite and quartz, Zone I; (**c**) Mineral association composed of cleavelandite (albite), muscovite, and lepidolite; (**d**) Beryl crystals embedded within a lepidolite envelopment at the contact between Zones III and IV; (**e**) Transition from schorl to blue-colored tourmaline; (**f**) Red-colored tourmaline from the Boqueirão granitic pegmatite.

## 3. Materials and Methods

A fluid inclusion study was carried out on microcline and quartz crystals collected from Zone II. Muscovite grains gathered from Zone I were suitable for $^{39}$Ar/$^{40}$Ar dating.

Petrographic and microthermometric measurements of fluid inclusions were performed at the University of Zagreb. Double polished, 0.1–0.3 mm thick, transparent mineral wafers were studied. Measurements were carried out on Linkam THMS 600 (Linkam Scientific Instruments Ltd., Tadworth, UK) stages mounted on an Olympus BX 51 (Olympus, Tokyo, Japan) using 10× and 50× Olympus long-working distance objectives. Two synthetic fluid inclusion standards (SYN FLINC; pure $H_2O$

and mixed $H_2O$–$CO_2$) were used to calibrate the equipment. The precision of the system was <2.0 °C for homogenization temperatures, and <0.2 °C in the temperature range between −60 and +10 °C. Microthermometric measurements were conducted on carefully defined fluid inclusion assemblages, representing groups of inclusions that were trapped simultaneously. The fluid inclusion assemblages were identified based on petrography prior to heating and freezing. If all of the fluid inclusions within the assemblage showed similar homogenization temperatures, the inclusions were assumed to have trapped the same fluid and to have not been modified by leakage or necking; these fluid inclusions thus record the original trapping conditions [20–22]. The salinity of aqueous inclusions was calculated from the final ice melting temperature using the BULK computer program [23]. Calculations are based on purely empirical best–fits, with no fundamental thermodynamic modeling involved [24]. The salinity of aqueous-carbonic inclusions was calculated according to [25]. Isochores were calculated by the ISOC computer program [23].

Raman spectroscopy of fluid inclusions, used for the semiquantitative analysis of entrapped volatiles, was performed on a Dilor LabRAM instrument (Horiba, Kyoto, Japan). Investigations were carried out at Department of Mineralogy and Petrology, Montanuniversität Leoben. A laser beam was focused through an Olympus BX 40 microscope (Olympus, Tokyo, Japan) onto the fluid inclusion of interest. The objective lenses of 50× and 100× magnification, combined with a confocal optical arrangement, enable a spatial resolution in the order of a cubic micrometer. A frequency-doubled Nd-YAG green laser (532 nm, 100 mW) was employed.

The $^{40}Ar/^{39}Ar$ analysis of muscovite was carried out at the ARGONAUT laboratory at the Department of Geography and Geology, Paris-Lodron-University of Salzburg. Mineral concentrates of muscovite were packed in aluminum-foil, sealed in quartz vials, and irradiated in the MTA KFKI reactor (Budapest, Hungary) for 16 h. The neutron fluence was monitored with DRA1 sanidine standard for which a $^{40}Ar/^{39}Ar$ plateau age of 25.03 ± 0.05 Ma has been reported [26]. Analyses were performed using a defocused (~1.5 mm diameter) 25 W $CO_2$-IR laser operating at the wavelengths between 10.57 and 10.63 μm. Gas cleanup was performed using two Zr–Al SAES getters (Milan, Italy). Ar-isotopes were measured on the VG ISOTECHTM NG3600 mass spectrometer (Isotopx, Middlewich, UK) on an axial electron multiplier in a static mode. Intensities of the peaks were back-extrapolated over 16 measured intensities to the time of gas admittance either with a straight line or a curved fit. Intensities were corrected for system blanks, background, post-irradiation decay of $^{37}Ar$, and interfering isotopes. Isotopic ratios, ages, and uncertainties for individual steps were calculated following the suggestions by [27] and [28] using decay factors reported by [29]. The calculation of the plateau age was carried out using ISOPLOT/EX [28].

To avoid $^{40}Ar/^{39}Ar$ dating of hydrothermally altered muscovite, prior to the $^{40}Ar/^{39}Ar$ analysis an aliquot of the muscovite sample was analyzed by applying the X-ray powder diffraction (XRD) technique. The XRD analysis was conducted at the University of Zagreb on a Philips PW 3040/60 X'Pert PRO powder diffractometer (45 kV, 40 μA), with CuK-monochromatized radiation (λ = 1.54056 Å) and θ–θ geometry. The area between 4 and 63° 2θ, with 0.02 steps, was measured with a 0.5 primary beam divergence.

## 4. Results

### 4.1. Petrography of Fluid Inclusions

Microscopic examinations, performed at the room temperature on double-side-polished microcline and quartz wafers from Zone II, distinguished four types of inclusions:

Type I. Two phase-aqueous fluid inclusions (FIs) are mostly irregular, but some of them show progressive formation of negative crystal forms. The degree of fill (F), around 0.9, is fairly uniform (Figure 4a–c).

Type II. Aqueous-carbonic FIs show mostly irregular forms. This type of inclusion is characterized by the presence of two immiscible liquid phases ($L_1$ and $L_2$) and a vapor (V) phase. The F value

varies slightly around 0.7 (Figure 4a,d). $L_1$ represents an aqueous solution whereas $L_2$ is composed of liquid $CO_2$.

Type III. Monophase elongated or slightly irregular gas inclusions (Figure 4e).

Type IV. Melt inclusions are mostly polyphase (Figure 4f), but they do not undergo any phase transition in the temperature range between −180 and +600 °C.

All four types of inclusions occur together in numerous inclusion assemblages, reflecting a melt-fluid immiscibility during crystallization of Zone II microcline and quartz.

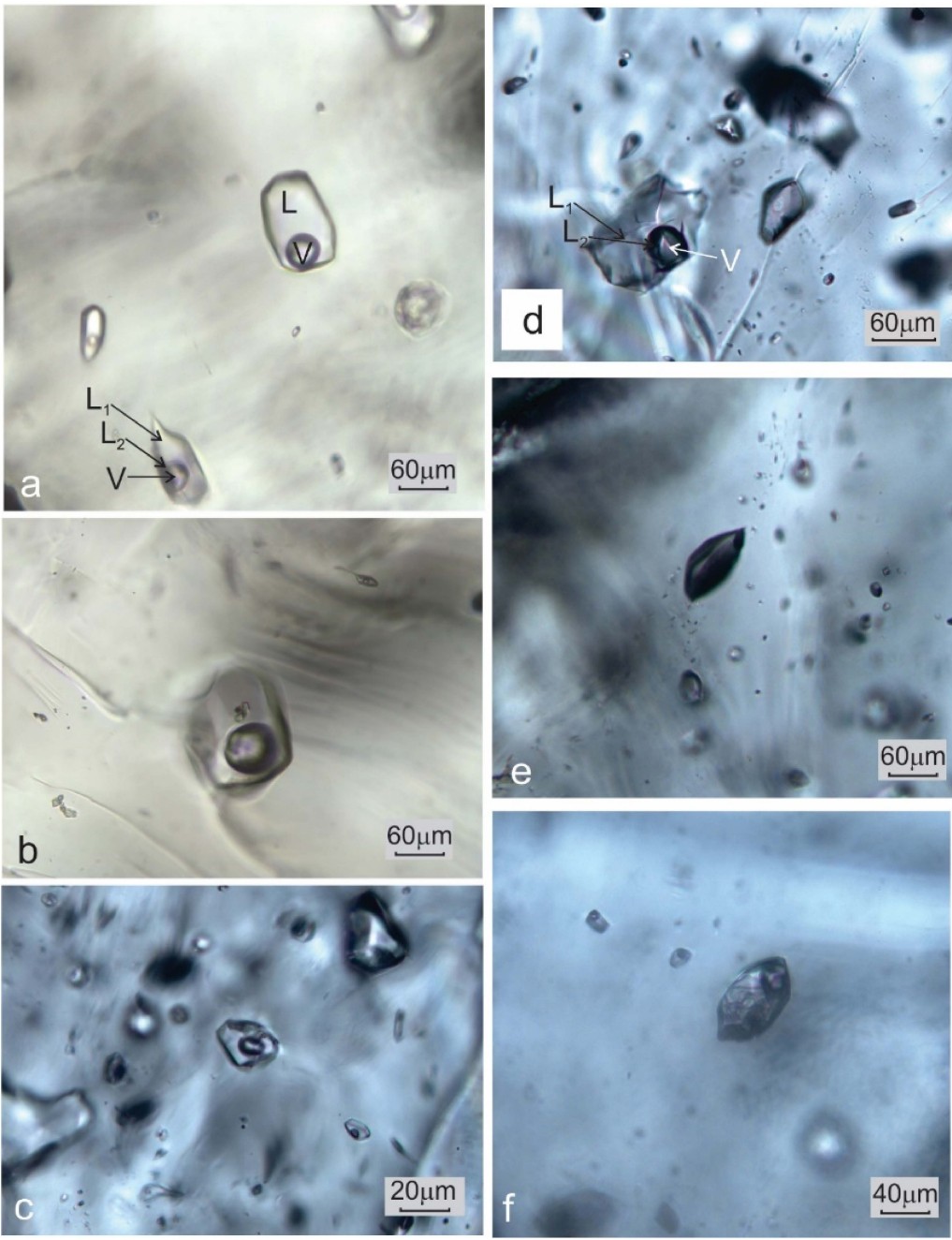

**Figure 4.** Macrophotographs of (**a**) coexisting two-phase, L + V, aqueous fluid inclusions and three-phase, $L_1$ + $L_2$ + V, aqueous-carbonic inclusions hosted by quartz; (**b**) two-phase, L + V, aqueous fluid inclusions seldom contain accidentally entrapped solid phases; (**c**) two-phase, L + V, aqueous fluid inclusions in microcline; (**d**) aqueous-carbonic inclusions in microcline; (**e**) a vapor-only inclusion hosted by microcline.

### 4.2. Microthermometry of Fluid Inclusions

Microthermometric data were collected from Type I and Type II inclusions hosted by microcline and quartz (Figure 5). Monophase fluid inclusions (Type III) as well as melt inclusions (Type IV) do not show any phase transition in the temperature range between −180 and +600 °C.

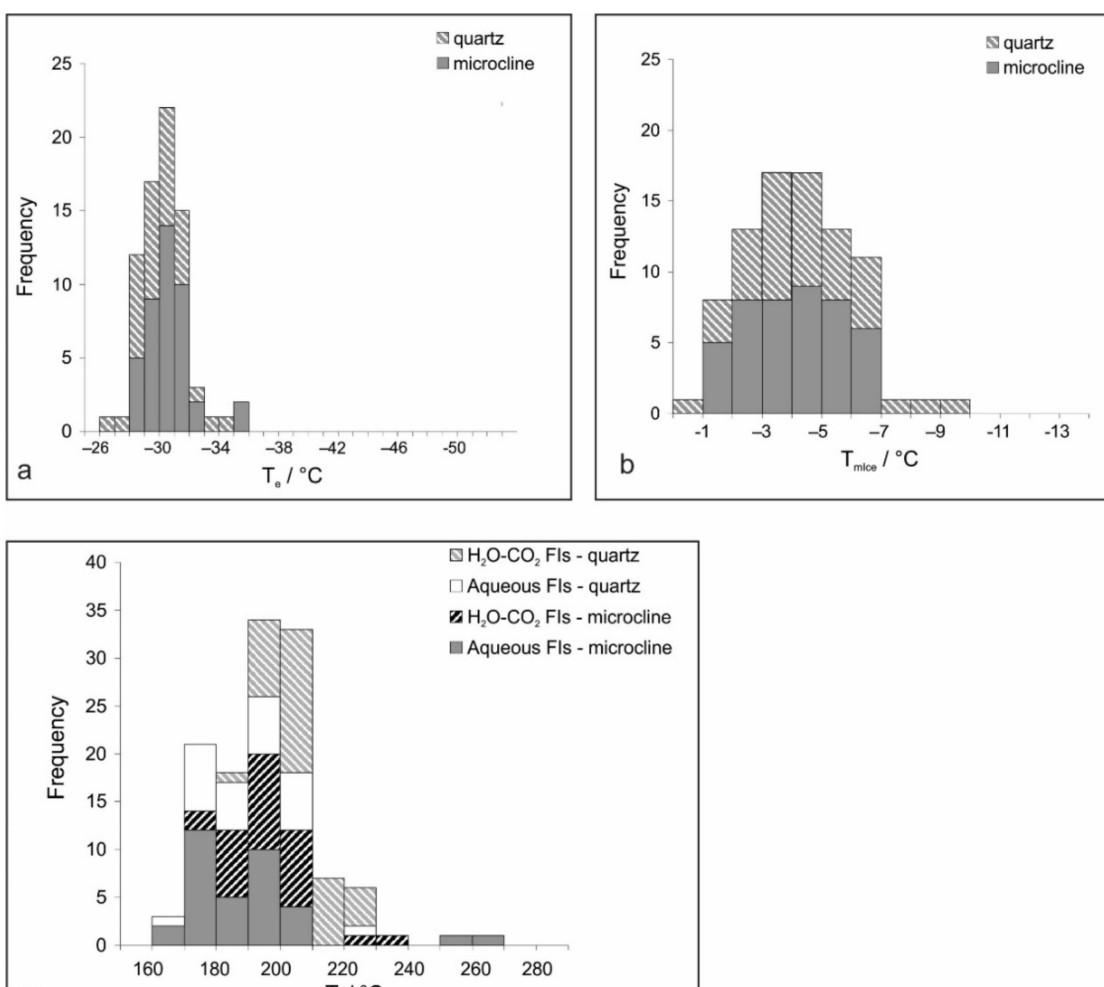

**Figure 5.** Histograms showing frequency distributions of fluid inclusion data: (**a**) Eutectic temperature ($Te$); (**b**) Final melting temperature of ice ($T_{mIce}$); (**c**) Homogenization temperature ($T_h$).

Microcline; Type I. A colorless frozen content of the aqueous FIs, formed at moderately low temperatures (generally around −40 °C), could be observed only by distortion and shrinkage of the vapor bubble. The initial melting temperature (eutectic temperature, $T_e$) is observed in an interval between −25.2 and −32.7 °C (Figure 5a) and the final ice melting ($T_{mIce}$) occurs between −1.8 and −6.2 °C (Figure 5b), reflecting the apparent salinity of 3.1–9.5 wt.% equ. NaCl. Homogenization ($T_h$) follows the disappearance of the vapor phase at 170–220 °C (Figure 5c). The bulk fluid density, calculated utilizing the equation of state proposed by [30] through the BULK software [23], spans from 0.867 to 0.965 g/cm$^3$. Type II carbonic-aqueous FIs are composed of a low-density gaseous phase and higher-density aqueous and carbonic liquid phases. The initial freezing of the aqueous phase was recorded around −40 °C, and complete freezing occurs at around −100 °C. The first melting of solid $CO_2$ occurs mostly in the temperature range between −60.5 and −67.8 °C, which is assigned to the presence of other volatiles (such as $CH_4$ and $N_2$; [31]). Melting of the aqueous part of inclusions was observed in a wide range of temperatures between −40 and −2 °C. The final melting of clathrate ($T_{mClath}$) spans between 8.0 and 9.0 °C and reflects salinities ranging from 2.5 to 4.0 wt.% equ. NaCl. Homogenization of the carbonic phase

proceeds in two ways, L + V→V and L + V→L. Critical phenomena have not been observed. The data gather around 29 °C in either way of homogenization. Total homogenization ($T_h$) into the liquid phase occurred in the interval between 170 and 230 °C (Figure 5c). Bulk fluid densities, calculated according to the equation of state from [32] revised by [33], fall in the range between 0.808 and 0.893 g/cm$^3$.

Quartz, Type I: Measurements performed on aqueous FIs within quartz samples yielded microthermometric data that overlap with those gathered from microcline (Figure 5a–c). In contrast, Type II (carbonic-aqueous) FIs in quartz show some differences compared to the same type of FIs hosted by microcline. The initial melting of $CO_2$ occurs in the interval between −57.1 and −65.0 °C. The recorded $T_{mClath}$ between 7.0 and 9.0 °C points to the salinity between 2.5 and 6.1 wt.% equ. NaCl. Homogenization of the carbonic phase proceeds mostly, at around 20 °C, into a liquid phase (Figure 5e). A critical phenomenon was observed only in one inclusion at +19.9 °C. The Th values have been recorded in an interval between 185 and 225 °C (Figure 5c).

### 4.3. Raman Spectroscopy of Fluid Inclusions

Raman spectroscopy measurements performed on aqueous FIs (Type I) within microcline and quartz samples recognized only the presence of water. The measurements of Type II FIs suggest $CO_2$ as the major non-$H_2O$ volatile component (Figure 6a). In addition, variable amounts of $N_2$ were detected (Figure 6b). However, in several cases, the measured peak areas were too small for reliable quantification. A simple formula based on Placzek's polarizability theory was applied to derive quantitative molar fractions of species present in FIs [34–38]. The molar fraction of $N_2$ ranges up to 9 mole % (Table 1). No difference between quartz and microcline samples was observed.

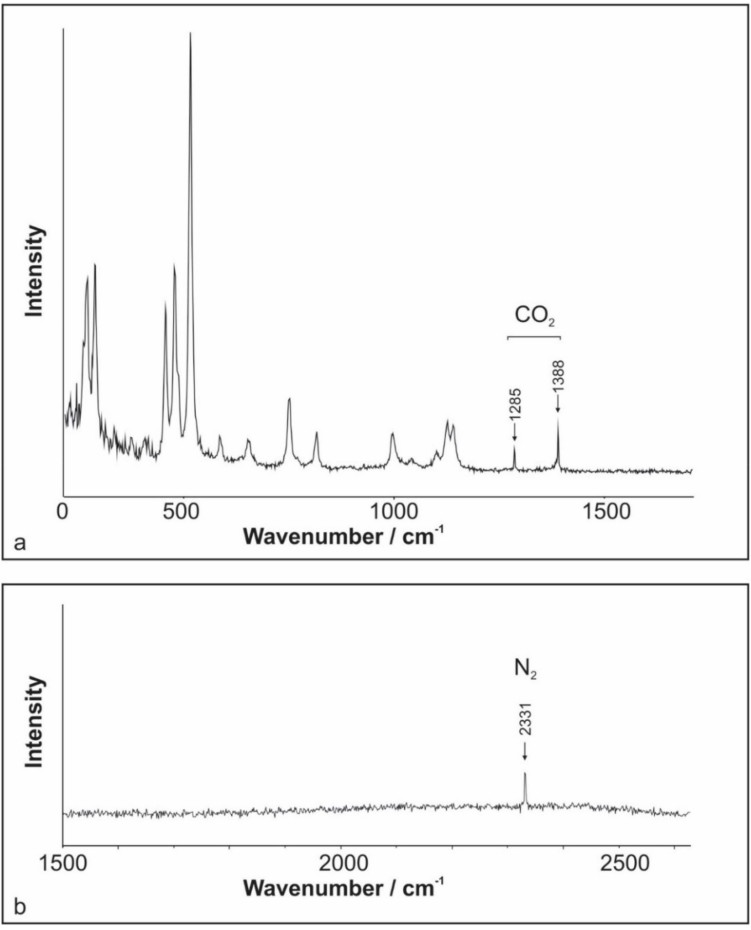

**Figure 6.** Raman spectra of the vapor bubble from an aqueous-carbonic inclusion reveal the presence of (**a**) $CO_2$ and (**b**) $N_2$.

**Table 1.** Microthermometric results, Raman data, and calculated bulk compositions of selected carbonic-aqueous fluid inclusions.

| Sample | FI | Microthermometry (°C) | | | | Raman (mol %) | | Bulk Composition (mol %) | | | Density (g·cm$^{-3}$) |
|---|---|---|---|---|---|---|---|---|---|---|---|
| | | $T_{mice}$ | $T_{mclath}$ | $T_{hCO2}$ | Mode | $CO_2$ | $N_2$ | $H_2O$ | $CO_2$ | $N_2$ | |
| | | | | | | Microcline | | | | | |
| m-1 | 1 | −2.5 | 8.1 | 26.2 | V | 93.2 | 6.8 | 85.4 | 11.6 | 0.7 | 0.9366 |
| m-1 | 2 | −3.1 | 8.6 | 28.3 | V | 96.5 | 3.5 | 84.4 | 12.6 | 0.2 | 0.9576 |
| m-1 | 3 | −4.3 | 7.8 | 27.5 | L | 98.0 | 2.0 | 83.4 | 12.6 | 0.1 | 0.9691 |
| m-1 | 4 | −2.0 | 8.9 | 27.9 | V | 94.7 | 5.3 | 86.7 | 10.9 | 0.6 | 0.9253 |
| m-2 | 1 | −2.7 | 8.7 | 30.0 | L | 99.5 | 0.5 | 84.3 | 13.2 | <0.05 | 0.9671 |
| m-2 | 2 | −3.3 | 9.2 | 28.0 | L | 97.8 | 2.2 | 84.2 | 12.7 | 0.2 | 0.9608 |
| m-3 | 1 | −4.0 | 8.1 | 28.6 | V | 92.1 | 7.9 | 83.6 | 12.1 | 0.6 | 0.9621 |
| m-3 | 2 | −2.9 | 7.6 | 29.1 | V | 99.9 | 0.1 | 84.1 | 13.2 | <0.01 | 0.9694 |
| m-3 | 3 | −3.1 | 8.8 | 28.8 | V | 93.3 | 6.7 | 85.3 | 11.5 | 0.4 | 0.9359 |
| | | | | | | Quartz | | | | | |
| q-1 | 1 | −4.0 | 8.2 | 20.1 | V | 93.4 | 6.6 | 83.7 | 12.0 | 0.8 | 0.9673 |
| q-1 | 2 | −3.1 | 8.9 | 18.5 | V | 99.7 | 0.3 | 82.8 | 14.4 | <0.05 | 0.9950 |
| q-1 | 3 | −2.9 | 8.4 | 17.9 | V | 92.6 | 7.4 | 83.5 | 13.0 | 0.9 | 0.9749 |

### 4.4. The $^{40}Ar/^{39}Ar$ Age

A concentrate of a few relatively large flakes (<0.5 mm) of white mica (muscovite) free of any inclusions was selected for dating. The XRD pattern of the analysed muscovite is presented in Figure 7. The experimental results of $^{40}Ar/^{39}Ar$ dating are given in Table 2. The argon release pattern of the muscovite sample (Figure 8) shows a slightly disturbed U-shaped pattern with a plateau age of 502.5 ± 5.8 Ma, constituting together 88.9% of $^{39}Ar$ released (Steps 4 to 10). Low-energy Steps 1 to 3 yielded an addition of excess argon, whereas a significant increase in some ages (Steps 6 and 8) is attributed to an internally inhomogeneous distribution of argon in muscovite. The $^{37}Ar_{Ca}$ values are low (Table 2) and the variation of $^{37}Ar_{Ca}$ is small. In contrast, the chlorine-derived $^{38}Ar_{Cl}$ values are relatively high (Table 2), showing that muscovite did grow under some saline conditions. We consider the plateau age of 502.5 ± 5.8 Ma to be geologically significant and to date the cooling of pegmatite through the argon retention temperature, which is experimentally determined at 425 ± 25 °C in slowly cooling terranes [39].

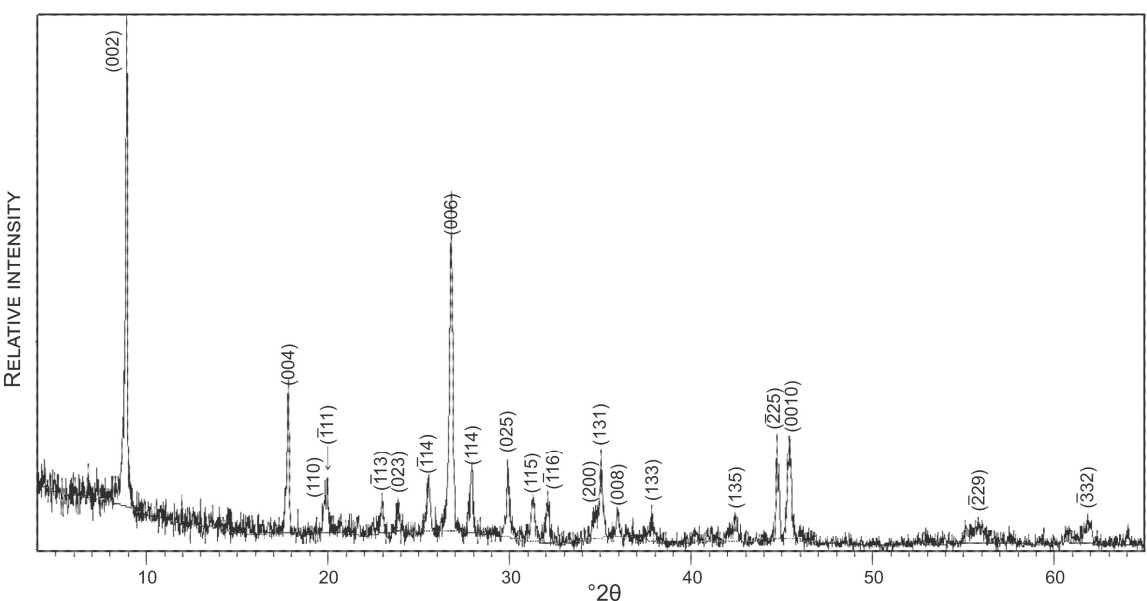

**Figure 7.** XRD pattern of muscovite dated by the of $^{40}Ar/^{39}Ar$ technique.

**Table 2.** The experimental results of $^{40}$Ar/$^{39}$Ar analytical data.

| J-Value: | | 0.01850 | +/− | 0.00019 | | | | | | | | | | | | |
|---|---|---|---|---|---|---|---|---|---|---|---|---|---|---|---|---|
| Step | $^{36}$Ar Meas. | $\pm\sigma_{36}$ | $^{37}$Ar | $\pm\sigma_{37}$ | $^{38}$Ar | $\pm\sigma_{38}$ | $^{39}$Ar | $\pm\sigma_{39}$ | $^{40}$Ar | $\pm\sigma_{40}$ | $^{40}$Ar*/$^{39}$Ar$_K$ | $\pm\sigma$ | %$^{40}$Ar* | %$^{39}$Ar | Age (Ma) | $\pm$(Ma) |
| | | | Decay Corr. | | Meas. | | Decay Corr. | | Meas. | | | | | | | 1-Sigma Abs. |
| 1 | 62.9 | 9.3 | 56 | 25 | 120 | 7.9 | 7820 | 66.0 | $1.96 \times 10^5$ | 215 | 22.69 | 0.40 | 90.5 | 2.0 | 632.4 | 10.9 |
| 2 | 24.7 | 7.2 | 55 | 19 | 120 | 13 | 4070 | 26.2 | $9.59 \times 10^4$ | 183 | 21.74 | 0.54 | 92.4 | 1.1 | 610.0 | 13.9 |
| 3 | 56.1 | 12 | 52 | 29 | 360 | 14 | 33,100 | 68.6 | $6.27 \times 10^5$ | 253 | 18.44 | 0.11 | 97.3 | 8.6 | 529.6 | 5.5 |
| 4 | 17.7 | 8.7 | 46 | 22 | 940 | 15 | 90,500 | 119 | $1.59 \times 10^6$ | 916 | 17.49 | 0.04 | 99.6 | 23.5 | 505.7 | 4.6 |
| 5 | 99.8 | 1.3 | 37 | 25 | 1100 | 13 | 111,000 | 22.0 | $1.95 \times 10^6$ | 1340 | 17.24 | 0.04 | 98.4 | 28.9 | 499.5 | 4.6 |
| 6 | 27.8 | 18 | 63 | 32 | 130 | 11 | 9500 | 68.3 | $1.83 \times 10^5$ | 274 | 18.36 | 0.58 | 95.5 | 2.5 | 527.5 | 15.3 |
| 7 | 17.2 | 7.0 | 10 | 25 | 1100 | 18 | 108,000 | 142 | $1.88 \times 10^6$ | 1630 | 17.28 | 0.03 | 99.7 | 28.2 | 500.5 | 4.6 |
| 8 | 37.5 | 15 | 67 | 27 | 170 | 11 | 12,300 | 53.1 | $2.39 \times 10^5$ | 289 | 18.48 | 0.38 | 95.3 | 3.2 | 530.5 | 10.6 |
| 9 | 28.8 | 11 | 120 | 32 | 260 | 14 | 23,300 | 54.9 | $4.24 \times 10^5$ | 338 | 17.85 | 0.14 | 98.0 | 6.1 | 514.7 | 5.9 |
| 10 | 51.1 | 13 | 120 | 39 | 120 | 13 | 6060 | 43.0 | $1.26 \times 10^5$ | 187 | 18.25 | 0.67 | 88.0 | 1.6 | 524.7 | 17.3 |
| Total | - | - | - | - | - | - | - | - | - | - | - | - | - | 100.0 | 506.3 | 4.6 |

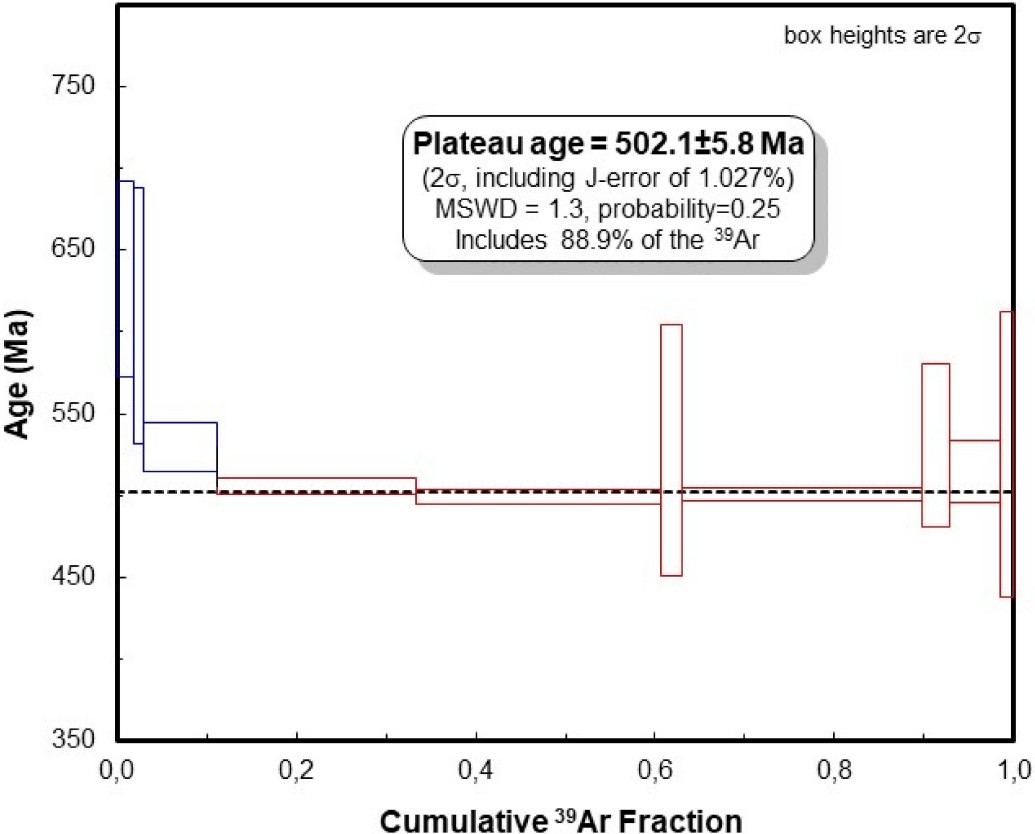

**Figure 8.** $^{40}Ar/^{39}Ar$ apparent age spectra of the coarse-grained muscovite sample. Laser energy increases from left to right. Vertical width of bars represents the $2\sigma$ error and includes the error of the J-value. Steps 4–10 are used for calculation of the plateau age.

## 5. Discussion

Fluid inclusions hosted by pegmatite crystals may provide a snapshot of P-T-X conditions at the time of their entrapment [40]. Numerous studies on the LCT-type of pegmatites worldwide have outlined the following types of inclusions: (1) melt inclusions; (2) saline aqueous fluid inclusions; and (3) $CO_2$-enriched fluid inclusions [41–45]. Classical theories of pegmatite genesis emphasize the importance of volatiles during crystallization of granitic pegmatites from coexisting aluminosilicate melt and hydrous fluids [46,47]. According to the model proposed by [48], aluminosilicate melts can produce highly evolved pegmatitic liquids via continuous crystallization under particular kinetic conditions. The ubiquitous crystallization commenced by formation of schorl (tourmaline), which buffered the boron content in the hydrous silicic melt, produces a high amount of exsolution of hydrous fluid [48]. The "boron quenching" in turn was enhanced by consequent crystallization of elbaite. The sink of boron, due to tourmaline crystallization and removal of the Li-alkali borate fluxing component, caused the separation of silicate and aqueous fluids producing supersaturation of alkali aluminosilicates (albite, microcline, quartz) and oxides, and their massive growth. Experimental studies support the coexistence of alumino-silicate melt, hydrous fluid, and hydrosaline fluid during the late stage of magma evolution [49].

Fluid inclusion assemblages, composed of melt inclusions and aqueous and $CO_2$-bearing fluid inclusions in minerals from Zone II of the gem-bearing Boqueirão granitic pegmatite, indicate the magmatic–hydrothermal transition. Similar phenomena have been recorded in other granitic pegmatites worldwide [50,51]. Melt inclusions represent entrapped remains of the silicate-rich melt, whereas fluid inclusions contain the fluid phase exsolved during the crystallization process. Additionally, coexistence of aqueous and aqueous-carbonic inclusions as well as their overlapping homogenization

temperatures reflect an immiscibility between the low-salinity, low-density $CO_2$-rich fluid phase and the higher-salinity, higher-density fluid phase during the magmatic–hydrothermal transition. The recorded three-phase immiscibility (silicate melt–low salinity and low density carbonic-aqueous fluid–moderate salinity and moderate density fluid) affected the fate of metals in the evolving pegmatite. The majority of lithophile metals stay in the silicate melt, but some elements preferentially enter immiscible fluids. The partitioning is strongly controlled by the salinity of the exsolving fluids [52]. Pb, Zn, Ag, and Fe preferentially partition into fluids with a higher salinity, whereas Mo, B, As, Sb, and Bi prefer low-salinity fluids. In contrast, Li and Sn do not show systematic variations in their partition coefficients with the salinity of the fluids [53]. Partitioning of P between silicate melts and exsolving fluids strongly depends on pressure and temperature. Regardless, partition of flux elements (e.g., B, Li, and P) into exsolving fluids together with the loss of $H_2O$ during the magmatic–hydrothermal transition may increase the viscosity of evolving silicate melts and affect textural features of pegmatites [54].

The lithium aluminosilicate phase diagram has been used as a petrogenetic grid for lithium-rich pegmatites [55]. Spodumene and petalite are stable Li-phases in a quartz-saturated system up to a temperature of 700 °C, which sets the upper limit on the crystallization conditions for Li-rich pegmatites [56]. However, the Boqueirão granitic pegmatite is characterized by the presence of spodumene as the only $LiAl(SiO_3)_2$ polymorph, which reflects the minimum formation pressure of 1.6 kbar (Figure 9).

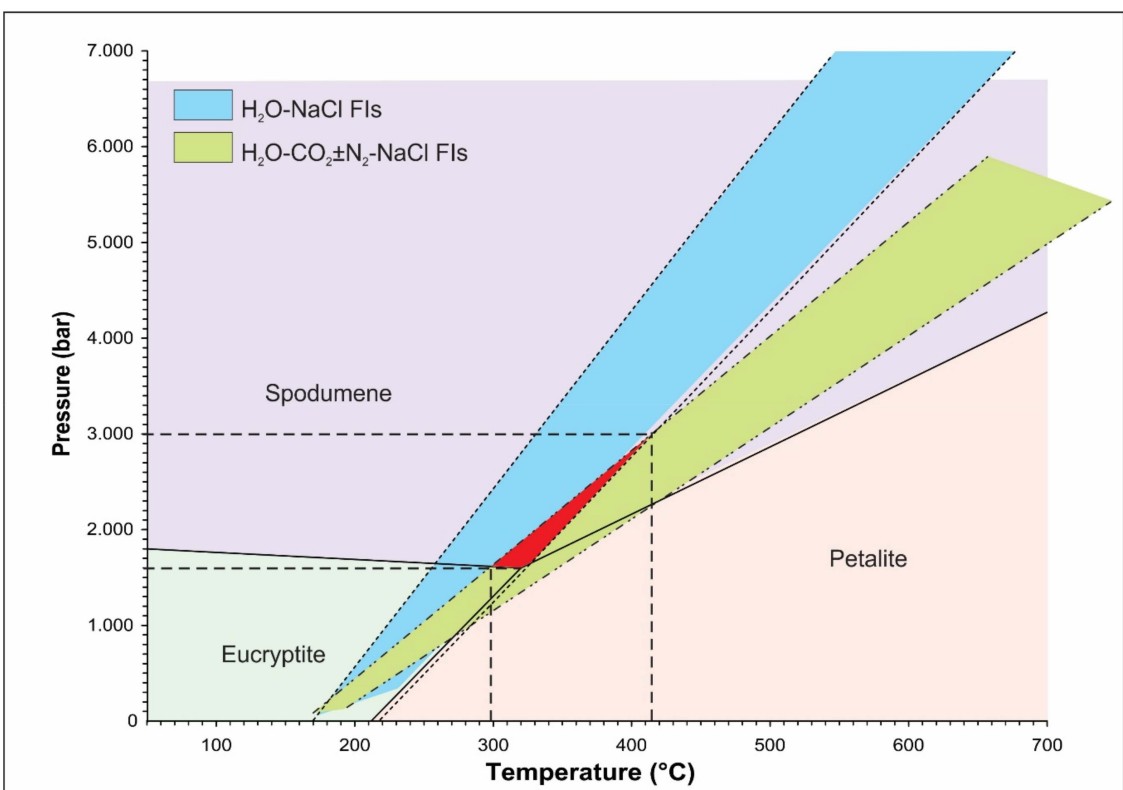

**Figure 9.** The isochores for aqueous ($H_2O$–NaCl FIs) and aqueous-carbonic ($H_2O$–$CO_2$ ± $N_2$-NaCl FIs) fluid inclusions extrapolated across the stability field of spodumene suggest a formation temperature in the range between 300 and 415 °C at a pressure from 1.8 to 3 kbar.

Isochores, constructed for the coexisting aqueous and aqueous-carbonic fluid inclusions (calculated using the equation of state proposed by [57] for the NaCl–$H_2O$ system and [32] revised by [33] for the $H_2O$–$CO_2$–$CH_4$–$N_2$–NaCl system) extrapolated across the stability field of spodumene suggest that the magmatic–hydrothermal transition associated with the formation of the Boqueirão granitic pegmatite occurred at a temperature from 300 to 415 °C and a pressure between 1.8 and 3 kbar (Figure 9).

According to the $^{40}Ar/^{39}Ar$ plateau age of muscovite, the Boqueirão granitic pegmatite crystallized simultaneously with late stage of magmatic activity in the BPP (511–500 Ma; e.g., [58]).

## 6. Conclusions

According to its mineralogical and geochemical characteristics, the Boqueirão granitic pegmatite has been classified as a member of the LCT pegmatite family, broadly widespread over the BPP. The pegmatite shows a zonal structure with significant enrichment on incompatible elements from its outer rim toward the massive quartz core.

Coexistence of melt inclusions and aqueous and $CO_2$-enriched fluid inclusions suggests the magmatic–hydrothermal transition that resulted with segregation of two liquid phases (low salinity and low density versus moderate salinity and moderate density) from the silicate melt. The fluid inclusion data together with the well-defined stability of $LiAl(SiO_3)_2$ polymorphs over the P-T area can be used as an indicator of formation conditions for this type of granitic pegmatite. The fluid inclusion data obtained from the Boqueirão granitic pegmatite accompanied by the P-T stability of spodumene revealed that the magmatic–hydrothermal transition occurred in the temperature range between 300 and 415 °C at a pressure ranging from 1.8 to 3 kbar.

The $^{40}Ar/^{39}Ar$ plateau age of muscovite, at 502.5 ± 5.8 Ma, sets the Boqueirão granitic pegmatite to the late stage of magmatic activity in the BPP.

**Author Contributions:** Conceptualization, S.S.P., L.P. and V.B.; Methodology, F.N., L.P., R.S. and S.S.P.; Formal Analysis, S.S.P. and S.B.Š.; Writing-Original Draft Preparation, all coauthors; Writing-Review & Editing, S.S.P., F.N. and R.S.; Visualization, S.S.P.; Funding Acquisition, S.S.P., L.P. and V.B.

**Funding:** This research was supported by the Croatian Ministry of Sciences, Technology, and Sports (Projects No. 119-0000000-1158 and 119-0982709-1175).

**Acknowledgments:** Early versions of this paper greatly benefited from comments by David London and Rainer Thomas. Ronald J. Bakker is appreciated for help in obtaining the Raman spectroscopy data. Special thanks go to Panagiotis Voudouris and Vasilios Melfos for handling the manuscript as well as to the anonymous reviewers whose comments substantially improved the quality of this paper. The publication charges for this article have been funded by a grant from the publication fund of UiT The Arctic University of Norway.

**Conflicts of Interest:** The authors declare no conflict of interest.

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
