# Peer review of "Formation Conditions and 40Ar/39Ar Age of the Gem-Bearing Boqueirão Granitic Pegmatite, Parelhas, Rio Grande do Norte, Brazil"

_minerals, doi:10.3390/min9040233_

Round 1
Reviewer 1 Report
I have read the manuscript by Palinkas et al. and I find the research welcome and an important addition to the dialogue of the pressure and temperature range for the formation of the Boqueirao pegmatites located in the Northeastern Brazil. The research represents a refinement of previous pressure and temperature estimates.
The paper seems to be well structured, with nice figures and of adequate length. It should be noted that I’m not qualified to critique the chosen methodologies and the language apart from the geochronology section, so my review focus on the research/outline/presentation, which gives some opportunities to improve the manuscript, as specified below. With some moderate changes, I find the manuscript suitable for inclusion in Minerals.
Minor points and comments
Introduction. I'd like to suggest that you make more of an effort to underscore the importance of this study in the Abstract and especially in the Introduction. Why was this study necessary? Why is this area important? Why is this approach better? What can we learn from this?
Refer to the 40Ar/39Ar method consistently throughout the manuscript.
Even though English is not my mother tongue, I feel that the manuscript needs a language wash. It has too many misuses of “the”, spelling mistakes and weird sentence constructions. It lacks the English style and elegancy.
The result chapter is a tough piece to read. I suggest a rewrite. Can you put the major findings in a table? Is it necessary to be so detailed? Can you refer to the methodology and simplify the text? Be more to the point.
I really think you should discuss your findings in the light of previous estimates. The geochronology result is not even discussed. I think you should lift the new findings up and its implications on a more regional scale.
I could not find a table with degassing results following figure 7.
Line specific points
Line 55-56. Why not convert their MPa range to kbar so it’s easier to compare with yours?
Line 113. (532 nm, 100mV), should it be 100mW (milliwatts)?
Line 119. It’s not the flux you are monitoring, it’s the neutron fluence – the time integrated flux.
Line 122. The wavelength is μm, not mm. Skip “the” in “the two”.
Line 128. It is McDougall & Harrison (1999). Also, they list a wide range of different criteria from many authors. Which one do you follow?
Line 129. You write “The definition …” What do you mean with definition? Also, you have one date reported, so “ages” is not correct. Why not just write “Age calculation was carried out using ISOPLOT/EX (Ludwig, 2001)?
Figures
Fig. 7. Age spectra. I would suggest adding K/Ca or Ca/K spectra above the age spectra, so users could see clearer what is actually degassing. In that way it could reveal what is causing the hump (step 6) and thereby argue not to include that step in the calculation, but the hump should be visible of course. Ca is calculated from 37Ar whereas K from 39Ar(k).
Fig. 8. Would it be better to label the axis like P (bar) and T (⁰C)?
Author Response
Minor points and comments
Introduction. I'd like to suggest that you make more of an effort to underscore the importance of this study in the Abstract and especially in the Introduction. Why was this study necessary? Why is this area important? Why is this approach better? What can we learn from this?
Thank you for this comment, in the revised text we have emphasized the importance of this study as well as the importance of the studied area.
Refer to the 40Ar/39Ar method consistently throughout the manuscript.
We have accepted this suggestion and referred the 40Ar/39Ar method consistently throughout the manuscript
Even though English is not my mother tongue, I feel that the manuscript needs a language wash. It has too many misuses of “the”, spelling mistakes and weird sentence constructions. It lacks the English style and elegancy.
The revised manuscript has been corrected by a native English speaker
The result chapter is a tough piece to read. I suggest a rewrite. Can you put the major findings in a table? Is it necessary to be so detailed? Can you refer to the methodology and simplify the text? Be more to the point.
We have rewritten the result chapter accordingly.
I really think you should discuss your findings in the light of previous estimates. The geochronology result is not even discussed. I think you should lift the new findings up and its implications on a more regional scale.
Thank you for this comment, we accepted it and we think that it contributed to a better quality of the text. We added a short discussion on the implications of the geochronological results. Note that we recalculated the age with a newer version of the computer program (Ludwig, 2012).
I could not find a table with degassing results following figure 7.
Table 2 with results of analytical data has been added.
Line specific points
Line 55-56. Why not convert their MPa range to kbar so it’s easier to compare with yours?
Thank you for the comment, it has been accepted.
Line 113. (532 nm, 100mV), should it be 100mW (milliwatts)?
Yes, the comment has been accepted.
Line 119. It’s not the flux you are monitoring, it’s the neutron fluence – the time integrated flux.
Accepted.
Line 122. The wavelength is μm, not mm. Skip “the” in “the two”.
Accepted.
Line 128. It is McDougall & Harrison (1999). Also, they list a wide range of different criteria from many authors. Which one do you follow?
We corrected that type of interpretation: it is a plateau age and includes the error of the J-value, too (according to Ludwig, 2012) .
Line 129. You write “The definition …” What do you mean with definition? Also, you have one date reported, so “ages” is not correct. Why not just write “Age calculation was carried out using ISOPLOT/EX (Ludwig, 2001)?
The reported age is a plateau age.
Figures
Fig. 7. Age spectra. I would suggest adding K/Ca or Ca/K spectra above the age spectra, so users could see clearer what is actually degassing. In that way it could reveal what is causing the hump (step 6) and thereby argue not to include that step in the calculation, but the hump should be visible of course. Ca is calculated from 37Ar whereas K from 39Ar(k).
The argon release pattern includes now the error of the J-value and represents a plateau. We We did not add the Ca/K ratio to the age spectrum but refer to the table with analytical results. The Ca/K ratio in white mica is so small and has no influence on age. Generally, the Ca/K ratio is useful for Ca-bearing minerals like feldspar and amphbole. The variation of 37ArCa is small and the hump is not created by a particularly higher 37ArCa value. In the hump, the 37ArCa /3)ArK is 0.00659 and not particularly high compared to the steps before and afterwards: 0.00034 to 0.00532.
Fig. 8. Would it be better to label the axis like P (bar) and T (⁰C)?
Yes, we have changed the labels.
Reviewer 2 Report
Dear authors,
I went through your interesting manuscript about the formation conditions and dating of the Boqueirao pegmatite.
The paper is well written, the presentation of the acquired data is well organized and I think this work is of high interest to a broad audience.
Attached you may find a pdf file, which contains my remarks on the text, which are mainly typos, along with minor changes I suggested.
It was my pleasure to review your interesting work.
With my best regards

Author Response
Thank you for your revision. All editorial comments have been accepted.
Reviewer 3 Report
The research is not conducted and presented correctly.
The Ar/Ar dating chapter is very short. The date is good but :
- Normally, all Ar-Ar study is accompanied by a detailed Table of the degazing stages with the calculated and integrated ages. This table is not presented and so we cannot follow the global calculation for etermining the age.
- The size of the grains had to be precised to the reader...whatever you use monograins....they had to be in the same size range if you have several grains.
- X-Ray analysis has not been done on the muscovite to see if the mineral was pure or partly affected by chlorite to explain that we have not a very flat plateau.
- The chemical composition of the micas is always of use for interpreting the Ar/Ar data.
The fluid inclusions (FI) study presents several flaws on different aspects:
The petrographic study is very short.
- There is no description about the type of quartz and the distribution of FI. There is no precision about the description and distribution of the three FI types and also to know if these FI are primary or secondary. This is a main point: primary all ? no trails with secondary FI in such quartz associated with pegmatites ?
- No
precision.When looking at the aspect of the quartz (Fig. 3d) there is
probably a lot of trails with secondary FI. I cannot imagine that all
the FI are primary.
- The melt inclusions are present but only
one sentence and one photograph. No study and so the whole FI study is
incomplete to propose a P-T evolution of the gem-bearing pegmatite.
Microthermometry: the research is not conducted correctly:
-
the eutectic temperatures do not indicate that the fluids were trapped
in the H2O-NaCl or H2O-CO2-NaCl system. With such eutectic temperatures
you can be in the H2O-NaCl-MgCl2 (T. eutectic = -35°C), H2O-MgCl2 (T.
eutectic = -33.6°C)...so you had to explain How you calculate the global
salinity?.
- in the global calculation of the composition of the fluid you do not consider the salt (page 9, Table 1). why ?
- lines 157-158: difficult to understand
- What are the TmCO2 (temperature of melting of the CO2 ice) ? Nothing is presented in the text and figures ??? With these data you can draw a TmCO2 vs. ThCO2 diagram to see the evolution of your CO2 phase.
Discussion
-
You have not proved that your FI are primary or secondary, the relation
of your FI trapping relatively to quartz generation; there are no
information about the melt inclusions.
It seems to me that you go very fast for building a model like that you wrote by comparing with other studies on pegmatites.
- diagram P-T: I am not convinced that you can use this system with your fluids.If not you had to explain why you use this system. Probably the isochores are not these one with such coomplicated system in your case that was simplified to H2O or H2O-CO2-NaCl system.
- line 259
" coexistence of melt inclusions with H2O and CO2 FI": To use such an
evidence you had to study the melt inclusions.
It is my opinion. You
can also add the main informations obtained on this type of inclusion
obtained by Thomas et al. (2011) on Borborema pegmatites.
The different recommendations are found in the attached pdf document

Author Response
The Ar/Ar dating chapter is very short. The date is good but :
We expanded the description in the revised text.
Normally, all Ar-Ar study is accompanied by a detailed Table of the degazing stages with the calculated and integrated ages. This table is not presented and so we cannot follow the global calculation for etermining the age.
We added the table with analytical data to the revised manuscript.
The size of the grains had to be precised to the reader...whatever you use monograins....they had to be in the same size range if you have several grains.
In the revised text we clarified that we have measured a concentrate of a few grains ranging in their size between 160 and 500 µm.
X-Ray analysis has not been done on the muscovite to see if the mineral was pure or partly affected by chlorite to explain that we have not a very flat plateau.
From the optical point of view, the concentrate was pure and few grains (ca. 105 - 6) with the same optical properties were selected for analysis. We describe now the pattern in detail, which reflects some excess argon potentially liberated from neighboring K-feldspar. Also we added the XRD pattern of the analyzed muscovite.
The chemical composition of the micas is always of use for interpreting the Ar/Ar data.
Unfortunately, due to the short revision time we are not able to organized the electron microprobe analysis.
There is no description about the type of quartz and the distribution of FI. There is no precision about the description and distribution of the three FI types and also to know if these FI are primary or secondary. This is a main point: primary all ? no trails with secondary FI in such quartz associated with pegmatites ?
No precision. When looking at the aspect of the quartz (Fig. 3d) there is probably a lot of trails with secondary FI. I cannot imagine that all the FI are primary.
Thank you for your comment. In the revised text we have emphasized that both quartz and microcline that have been analyzed in this study are sampled from Zone II. Also the petrography of fluid inclusions have been extended. We applied the fluid inclusion assemblage approach suggested by Goldstein & Reynolds (1994), Goldstein (2001) and Bodnar (2003) rather than the classical subdivision to primary and secondary inclusions. We have added the explanation to the Samples and methods chapter.
The melt inclusions are present but only one sentence and one photograph. No study and so the whole FI study is incomplete to propose a P-T evolution of the gem-bearing pegmatite.
Whereas the melt inclusions do not show any phase transitions in the temperature range between -180 and +600°C, we can any use the presence of melt inclusions in fluid inclusion assemblages composed of melt and fluid inclusions as an indicator of the magmatic-hydrothermal transition.
The eutectic temperatures do not indicate that the fluids were trapped in the H2O-NaCl or H2O-CO2-NaCl system. With such eutectic temperatures you can be in the H2O-NaCl-MgCl2 (T. eutectic = -35°C), H2O-MgCl2 (T. eutectic = -33.6°C)...so you had to explain How you calculate the global salinity?.
In the global calculation of the composition of the fluid you do not consider the salt (page 9, Table 1). why ?
The apparent salinities have been expressed as wt.% NaCl equ. This approach does not require calculation of absolute concentrations of all dissolved salts (e.g., Roedder, 1984)
What are the TmCO2 (temperature of melting of the CO2 ice) ? Nothing is presented in the text and figures ??? With these data you can draw a TmCO2 vs. ThCO2 diagram to see the evolution of your CO2 phase.
TmCO2 represent the first melting of solid CO2.
Diagram P-T: I am not convinced that you can use this system with your fluids.If not you had to explain why you use this system. Probably the isochores are not these one with such coomplicated system in your case that was simplified to H2O or H2O-CO2-NaCl system.
Whereas we do not have LA-ICP-MS data obtained from individual fluid inclusions, we presented the fluid inclusions in two systems: H2O-NaCl and H2O-CO2-NaCl. Anyhow, we recalculated isochores using the different salts (MgCl2, CaCl2 and KCl) instead NaCl but slopes of isochrones in MgCl2-H2O, CaCl2-H2O or KCl-H2O systems did not vary significantly from those in H2O-NaCl system.
It is my opinion. You can also add the main informations obtained on this type of inclusion obtained by Thomas et al. (2011) on Borborema pegmatites.
Yes, the main findings are listed in the introduction chapter.
Reviewer 4 Report
Minor grammar and wording changes.
See attached PDF

Author Response
Thank you for your comments. We have accepted all editorial comments and also the revised manuscript has been corrected by a native English speaker.
Round 2
Reviewer 3 Report
nothing to add